# Residual Tumor Resection After Anti-PD-1 Therapy: A Promising Treatment Strategy for Overcoming Immune Evasive Phenotype Induced by Anti-PD-1 Therapy in Gastric Cancer

**DOI:** 10.3390/cells14151212

**Published:** 2025-08-06

**Authors:** Hajime Matsuida, Kosaku Mimura, Shotaro Nakajima, Katsuharu Saito, Sohei Hayashishita, Chiaki Takiguchi, Azuma Nirei, Tomohiro Kikuchi, Hiroyuki Hanayama, Hirokazu Okayama, Motonobu Saito, Tomoyuki Momma, Zenichiro Saze, Koji Kono

**Affiliations:** 1Department of Gastrointestinal Tract Surgery, Fukushima Medical University School of Medicine, 1 Hikarigaoka, Fukushima 960-1295, Japan; matsuida@fmu.ac.jp (H.M.); shotaro@fmu.ac.jp (S.N.); k-yamame@fmu.ac.jp (K.S.); s-h-0505@fmu.ac.jp (S.H.); yochi321@fmu.ac.jp (C.T.); azuma201@fmu.ac.jp (A.N.); tkiku@fmu.ac.jp (T.K.); hanayama@fmu.ac.jp (H.H.); okayama@fmu.ac.jp (H.O.); moto@fmu.ac.jp (M.S.); tmomma@fmu.ac.jp (T.M.); z-saze@fmu.ac.jp (Z.S.); kojikono@fmu.ac.jp (K.K.); 2Department of Blood Transfusion and Transplantation Immunology, Fukushima Medical University School of Medicine, 1 Hikarigaoka, Fukushima 960-1295, Japan

**Keywords:** anti-PD-1 therapy, gastric/gastroesophageal cancer, tumor resection, tumor immune microenvironment, HLA class I, TGF-β

## Abstract

Background: Anti-programmed death 1 receptor (PD-1) therapy is a promising treatment strategy for patients with unresectable advanced or recurrent gastric/gastroesophageal junction (G/GEJ) cancer. However, its response rate and survival benefits are still limited; an immunological analysis of the residual tumor after anti-PD-1 therapy would be important. Methods: We evaluated the clinical efficacy of tumor resection (TR) after chemotherapy or anti-PD-1 therapy in patients with unresectable advanced or recurrent G/GEJ cancer and analyzed the immune status of tumor microenvironment (TME) by immunohistochemistry using their surgically resected specimens. Results: Patients treated with TR after anti-PD-1 therapy had significantly longer survival compared to those treated with chemotherapy and anti-PD-1 therapy alone. Expression of human leukocyte antigen (HLA) class I and major histocompatibility complex (MHC) class II on tumor cells was markedly downregulated after anti-PD-1 therapy compared to chemotherapy. Furthermore, the downregulation of HLA class I may be associated with the activation of transforming growth factor-β signaling pathway in the TME. Conclusions: Immune escape from cytotoxic T lymphocytes may be induced in the TME in patients with unresectable advanced or recurrent G/GEJ cancer after anti-PD-1 therapy due to the downregulation of HLA class I and MHC class II expression on tumor cells. TR may be a promising treatment strategy for these patients when TR is feasible after anti-PD-1 therapy.

## 1. Introduction

Gastric cancer is the fifth most common cancer in terms of incidence and mortality worldwide [1]. For patients with unresectable advanced or recurrent gastric/gastroesophageal junction (G/GEJ) cancer, a combination therapy of chemotherapy with anti-programmed death 1 receptor (PD-1) therapy is the standard first-line treatment [2,3,4,5,6,7]. However, since the median overall survival (OS) of Stage IV G/GEJ cancer patients treated with this combination therapy was approximately 17–18 months [4,7], further improvements in treatment strategies are needed.

Conversion surgery is defined as a surgical treatment with the goal of R0 resection in initially unresectable gastric cancer patients after response to chemotherapy [8]. With advances in chemotherapy for unresectable advanced or recurrent G/GEJ cancer, a minor population of patients could undergo conversion surgery after a good response to chemotherapy, and it was reported that these patients had a better long-term prognosis [9,10]. The CONVO-GC-1 study reported that patients with Stage IV gastric cancer who underwent R0 resection after a favorable response to chemotherapy were expected to have long-term survival, indicating that conversion surgery may be a promising treatment strategy for them [11]. However, it remains unclear in daily clinical practice whether continuing chemotherapy or surgery is the better treatment strategy after a good response to chemotherapy in patients with unresectable advanced or recurrent G/GEJ cancer. These unmet clinical questions prompt us to evaluate biological and immunological analysis for residual tumors after response to chemotherapy.

Currently, anti-PD-1 therapy, immunotherapy, using nivolumab or pembrolizumab, is widely available in daily clinical practice for the treatment of advanced gastric cancer. In the Checkmate 649 trial, the combination of nivolumab and chemotherapy demonstrated superior efficacy compared with chemotherapy alone, with an objective response rate (ORR) of 60% versus 45% and a median OS of 14.4 months versus 11.1 months [3]. Similarly, in the KEYNOTE-859 trial, pembrolizumab combined with chemotherapy resulted in an improved ORR (51% vs. 42%) and median OS (12.9 vs. 11.5 months) compared to chemotherapy alone [5]. Although anti-PD-1 therapy is a promising treatment strategy for advanced gastric cancer patients, the response rate is still limited. To improve the efficacy of anti-PD-1 therapy, it is crucial to clarify the mechanism of resistance to anti-PD-1 therapy. The characteristics of the tumor immune microenvironment (TIME) after anti-PD-1 therapy may contribute to elucidating the mechanisms of anti-PD-1 therapy resistance and the selection of treatment options after anti-PD-1 therapy. However, the characteristics of TIME after anti-PD-1 therapy are not yet fully understood in gastric cancer.

In this study, we evaluated the efficacy of tumor resection (TR) after chemotherapy and anti-PD-1 therapy in patients with unresectable advanced or recurrent G/GEJ cancer initially. We also investigated the characteristics of TIME by immunohistochemistry (IHC) analysis using their surgically resected specimens obtained after chemotherapy or anti-PD-1 therapy. In addition, we investigated the mechanism of downregulation of human leukocyte antigen (HLA) class I expression on tumor cells from the viewpoint of transforming growth factor (TGF)-β signaling pathway using those surgically resected specimens.

## 2. Materials and Methods

### 2.1. Patients

The eligibility criteria for this study were as follows: (1) aged 20 years or older, (2) G/GEJ cancer with distant metastases diagnosed by computed tomography (CT), magnetic resonance imaging, ^18^F-fluorodeoxyglucose positron emission tomography/CT, or staging laparoscopy, (3) pathological type was adenocarcinoma, (4) first-line treatment with chemotherapy and/or anti-PD-1 therapy (immunotherapy) was administered at Fukushima Medical University Hospital between January 2017 and December 2023, (5) eastern cooperative oncology group performance status (ECOG-PS): 0–2. There are no exclusion criteria in this study.

We retrospectively enrolled 97 patients with unresectable advanced or recurrent G/GEJ cancer. The following clinical parameters were collected from clinical records: age, sex, ECOG-PS, primary tumor site (stomach or gastroesophageal junction), histological type (diffuse or non-diffuse), human epidermal growth factor receptor 2 (HER2) status, disease status (unresectable or recurrent), and non-curative factors (liver, peritoneum, distant lymph nodes, lung, and bone metastases, positive for peritoneal cytology, and tumor invasion of adjacent structures), the number of non-curative factors, the number of chemotherapy regimens, anti-PD-1 therapy, TR after chemotherapy or anti-PD-1 therapy, and laboratory data at first-line chemotherapy and/or anti-PD-1 therapy including serum albumin (Alb), neutrophil-to-lymphocyte ratio (NLR), alkaline phosphatase (ALP), creatinine (Cre), carcinoembryonic antigen (CEA), and carbohydrate antigen 19-9 (CA19-9). OS was defined as the period from the date of start of first-line treatment with chemotherapy and/or anti-PD-1 therapy (immunotherapy) to death from any cause or the date of last follow-up.

In 15 patients who underwent TR with curative intent after chemotherapy or anti-PD-1 therapy, the TIME was assessed by IHC using surgically resected specimens. The following clinical data were collected for these patients: pathological diagnosis, residual tumor status, pathological response to therapy, and clinical course before and after TR.

This study was approved by the Ethics Committee of Fukushima Medical University (approval number: REC2024-152) and was conducted in accordance with the principles of the Declaration of Helsinki.

### 2.2. IHC Staining

Paraffin-embedded sections (4 μm thick) were deparaffinized, rehydrated, and treated with 0.3% hydrogen peroxide to block endogenous peroxidase activity. Antigen retrieval and primary antibody staining were performed under conditions described in Appendix A. Detection was conducted using HRP-conjugated anti-rabbit or anti-mouse polymers (Envision+ System-HRP, K4003 or K001; Agilent Technologies, Santa Clara, CA, USA). Visualization was achieved with diaminobenzidine (Dojindo Molecular Technologies, Kumamoto, Japan), followed by counterstaining with Giemsa’s stain solution (Muto Pure Chemicals, Tokyo, Japan).

### 2.3. Assessment of IHC Staining

Programmed death ligand 1 (PD-L1) and PD-L2 expression levels were evaluated using the Combined Positive Score (CPS), while CD155, carcinoembryonic antigen-related adhesion molecule-1 (CEACAM-1), major histocompatibility complex (MHC) class II, and HLA class I were assessed using the H-score method. CD8 staining was assessed based on the number of stained lymphocytes, and T cell immunoglobulin and mucin domain-3 (TIM-3) expression was evaluated by the number of stained immune cells. T cell immunoglobulin and ITIM domain (TIGIT) was considered positive when there were lymphocytes with membranous staining. Phospho-Smad3 (p-Smad3) expression was evaluated as the percentage of tumor cells with nuclear staining. In this study, we identified tumor lesions by referring to hematoxylin-eosin stained sections of consecutive tissue sections and then evaluated each staining on tumor cells. Three authors (H.M., K.M., and S.N.), blinded to clinical data, independently evaluated the staining. For each slide, the tumor area was divided into five equal parts, and one field was randomly selected from each part. The mean value from these five fields was then calculated for each case.

We analyzed the correlation between the percentage of p-Smad3-positive tumor cells and the H-score of HLA class I expression on tumor cells within a total of 60 randomly selected identical fields (five fields per case, 12 cases in total). The expression of HLA class I was compared between two groups: one group with <5% of p-Smad3-positive tumor cells (30 fields) and the other group with ≥5% of those (30 fields). The expressions of p-Smad3 and HLA class I were evaluated using consecutive tissue sections.

### 2.4. Statistical Analysis

All statistical analyses were performed using the SPSS software program for Windows (version 30; IBM, Armonk, NY, USA). Continuous variables were compared using Welch’s *t*-test, and categorical variables were compared using the chi-square test or Fisher’s exact probability test. To analyze the prognostic factors, factors that were significant in univariate analysis using the log-rank test were subjected to multivariate analysis using Cox proportional hazards regression models. OS was analyzed using the Kaplan–Meier method, and comparisons between survival curves were evaluated using the log-rank test. In all analyses, statistical significance was set at *p* < 0.05.

## 3. Results

In this study, we evaluated the therapeutic potential of TR after chemotherapy and anti-PD-1 therapy (immunotherapy) from the clinical (Section 3.1, Section 3.2, Section 3.3 and Section 3.4) and TIME (Section 3.5 and Section 3.6) perspectives.

### 3.1. Characteristics of Patients Treated with Chemotherapy and Anti-PD-1 Therapy

The clinical characteristics of 97 patients with unresectable advanced or recurrent G/GEJ cancer treated with chemotherapy and/or anti-PD-1 therapy are shown in Table 1. These 97 patients comprised 70 male and 27 female patients with a mean age of 67.3 years (Table 1). In total, 82 patients underwent chemotherapy and/or anti-PD-1 therapy alone (Non-TR group), and 15 patients underwent TR with curative extent after chemotherapy or anti-PD-1 therapy (TR group). The TR group had significantly higher Alb levels and significantly lower ALP levels compared to the Non-TR group at the initiation of chemotherapy (*p* = 0.032 and *p* = 0.010, respectively) (Table 1). No significant differences were observed in other baseline characteristics between the two groups (Table 1). Additionally, there were no significant differences in non-curative factors between the two groups (Appendix A).

### 3.2. Prognostic Factors

Univariate and multivariate analyses of OS were performed using clinical characteristics and clinical laboratory data at first-line treatment with chemotherapy and/or anti-PD-1 therapy (immunotherapy) for all patients (Table 2). In the univariate analysis, HER2 positive, TR after chemotherapy or anti-PD-1 therapy, Alb ≥ 3.6, and NLR < 3.0 were significantly associated with a better OS (Table 2). Multivariate analysis was performed for the four variables that showed significant differences in the univariate analysis, and HER2 positive (*p* = 0.046; HR 0.408; 95% CI, 0.169–0.986) and TR after chemotherapy or anti-PD-1 therapy (*p* = 0.003; HR 0.274; 95% CI, 0.117–0.643) were significant independent predictors of favorable OS (Table 2).

### 3.3. Characteristics of Patients in the TR After Chemotherapy or Anti-PD-1 Therapy

The clinical course of 15 patients in the TR after chemotherapy or anti-PD-1 therapy is shown in Figure 1 and Figure 2. These patients were diagnosed as eligible for curative resection after chemotherapy or anti-PD-1 therapy and underwent TR at Fukushima Medical University Hospital. They were divided into two groups: nine patients who underwent TR after chemotherapy (Chemo+TR group) and six patients who underwent TR after anti-PD-1 therapy (Anti-PD-1+TR group). Case 14 underwent TR just after approximately one year of chemotherapy following anti-PD-1 therapy and was therefore included in the Chemo+TR group.

The characteristics of these 15 patients are shown in Table 3. These patients comprised 12 male and 3 female patients with a mean age of 64.2 years, and all patients had an ECOG-PS of 0. The Anti-PD-1+TR group was significantly older than the Chemo+TR group (*p* = 0.018) (Table 3). No patients in the Chemo+TR group had two or more non-curative factors, whereas three patients in the Anti-PD-1+TR group had two or more non-curative factors (*p* = 0.044) (Table 3). No patients in the Chemo+TR group had distant lymph node metastasis, whereas three patients in the Anti-PD-1+TR group had distant lymph node metastasis (*p* = 0.044) (Appendix A). There were no significant differences between the two groups regarding other non-curative factors. Overall, five patients (33.3%) were pathologically diagnosed with positive residual tumors (R1 or R2) (Table 3). A pathological complete response (Grade 3) was observed in two patients in the Chemo+TR group and one patient in the Anti-PD-1+TR group (Table 3).

### 3.4. OS for Each Treatment Group

The 5-year OS rate for all 97 patients was 15.7% (median OS 21.2 months), with a median follow-up period of 20.5 months (range 1.8–87.5 months). The 5-year OS rates were 52.2% (median OS not reached) in the TR group (15 patients) and 8.5% (median OS 19.2 months) in the Non-TR group (82 patients), and the TR group had a significantly longer OS than the Non-TR group (*p* < 0.001) (Figure 3a). There were 63 patients in the Non-TR group who underwent anti-PD-1 therapy (Anti-PD-1 group) (Table 1). In addition, the 5-year OS rates were 62.5% (median OS not reached) in the Anti-PD-1+TR group (6 patients), and 3.8% (median OS: 20.4 months) in the Anti-PD-1 group (63 patients), and the Anti-PD-1+TR group had a significantly longer OS compared to the Anti-PD-1 group (*p* = 0.022) (Figure 3b). There was no significant difference in OS between the Anti-PD-1+TR group (six patients) and the Chemo+TR group (nine patients) (Figure 3c).

### 3.5. Characteristics of TIME in Unresectable Advanced or Recurrent G/GEJ Cancer Treated with Chemotherapy or Immunotherapy

We evaluated the characteristics of TIME after chemotherapy or immunotherapy. In this study, three patients with pathological complete response (Grade 3) in the TR group were excluded from this analysis because tumor cell evaluation was not possible. We compared the TIME using IHC staining of surgical resection specimens between the Chemo+TR group (*n* = 7) and the Anti-PD-1+TR group (*n* = 5). Representative IHC staining images for HLA class I, CD8, TIM-3, TIGIT, PD-L1, PD-L2, CEACAM-1, CD155, and MHC class II in both groups were shown in Figure 4. HLA class I expression was significantly downregulated in the Anti-PD-1+TR group compared to the Chemo+TR group (*p* = 0.018) (Figure 5a). No significant differences were observed between the two groups in the number of CD8-positive T cells or TIM-3-positive infiltrating immune cells (Figure 5b,c). TIGIT expression was positive in only 2 cases in the Anti-PD-1+TR group but not in the Chemo+TR group (Appendix A). Among ligands for PD-1, PD-L1 expression was significantly upregulated in the Anti-PD-1+TR group compared to the Chemo+TR group (*p* = 0.025), whereas PD-L2 expression showed no significant difference (Figure 5d,e). MHC class II expression, a ligand for lymphocyte activation gene 3(LAG-3), was significantly downregulated in the Anti-PD-1+TR group (*p* = 0.018) (Figure 5f).

### 3.6. Downregulation of HLA Class I Expression in Relation to p-Smad3 on the Tumor Cells

We analyzed the expression level of HLA class I on tumor cells in relation to p-Smad3 expression levels using serial tissue sections. Representative IHC staining images for p-Smad3 and HLA class I expressions in serial tissue sections are shown in Figure 6a. Compared to the group with <5% of p-Smad3-positive tumor cells (30 fields), the group with ≥5% (30 fields) showed significantly downregulated HLA class I expression on tumor cells (*p* = 0.024) (Figure 6b). The difference was not statistically significant between p-Smad3-positive tumor cells in the Anti-PD-1+TR group and the Chemo+TR group (Figure 6c).

## 4. Discussion

This study indicated the following two findings in patients with unresectable advanced or recurrent G/GEJ cancer who treated chemotherapy and/or anti-PD-1 therapy (immunotherapy): (1) The Anti-PD-1+TR group had significantly longer survival compared to the Anti-PD-1 group; and (2) in comparison to tumor cells after chemotherapy, tumor cells after anti-PD-1 therapy showed significantly downregulated expression of HLA class I, which may be associated with the activation of TGF-β pathway. These findings suggest that TR may be a promising treatment strategy when TR is feasible after anti-PD-1 therapy.

When immunotherapy was performed, a previous report showed that immune escape mechanisms, such as adaptive immune resistance and acquired resistance, are induced in the tumor microenvironment [12]. Among both immune escape mechanisms, the following immune escapes may exist in anti-PD-1 therapy-refractory lesions: (1) loss or downregulation of HLA class I expression on tumor cells, (2) loss or downregulation of tumor antigens expression in tumor cells, (3) reduction in the number of tumor-infiltrating cytotoxic T lymphocytes(CTLs) and suppression of their function, (4) upregulated expression of immune checkpoint pathways other than the PD-1 axis, (5) infiltration of immunosuppressive cells into the TIME, (6) production of immunosuppressive cytokines such as TGF-β, and (7) abnormalities in the interferon (IFN)-γ signaling pathway in tumor cells, etc. [13,14].

Since CTLs recognize and eliminate tumor cells through a complex of tumor antigen-derived peptide and HLA class I molecule on the tumor cells, tumor cells escape from CTLs by losing or downregulating their HLA class I expression. The downregulation of HLA class I expression was reported as a component of the immune escape mechanisms associated with anti-PD-1 therapy [13]. In this study, HLA class I expression on tumor cells was significantly downregulated after anti-PD-1 therapy compared to chemotherapy (Figure 5a). In patients with unresectable advanced or recurrent G/GEJ cancer, immune escape due to downregulated HLA class I expression on tumor cells may be induced after anti-PD-1 therapy. Moreover, we found that HLA class I expression on tumor cells was significantly downregulated in tumor regions with a higher proportion of p-Smad3-positive tumor cells (Figure 6a,b). p-Smad3 is a key molecule in the TGF-β signaling pathway [15], and previous studies showed that TGF-β in the tumor microenvironment downregulates HLA class I expression on tumor cells of malignant melanoma [13,14]. Based on our findings and prior reports, it is suggested that TGF-β may also downregulate HLA class I expression on tumor cells in unresectable advanced or recurrent G/GEJ cancer. Although the difference was not statistically significant, there was a trend toward an increased frequency of p-Smad3-positive tumor cells in the Anti-PD-1+TR group compared to the Chemo+TR group (*p* = 0.085) (Figure 6c). While further investigation with an expanded sample size is needed, these findings suggest that anti-PD-1 therapy may induce downregulation of HLA class I expression on tumor cells through the TGF-β pathway in the TIME of unresectable advanced or recurrent G/GEJ cancer.

In addition to the PD-1 axis, it was well known that several immune checkpoint pathways, such as the TIGIT, TIM-3, and LAG-3 axes, suppress T-cell activity in TIME [16,17]. In this study, PD-L1 expression in the TIME was significantly increased after anti-PD-1 therapy compared to chemotherapy (Figure 5d). Previous studies reported that IFN-γ secreted by activated CD8-positive T cells after anti-PD-1 therapy acts on both tumor cells and immune cells in the surrounding TIME, thereby inducing their PD-L1 expression [18]. This situation is considered a mechanism of adaptive immune resistance and represents one of the factors that suppress antitumor immune responses. In our cohort, increased PD-L1 expression on tumor cells and immune cells in the TIME in the Anti-PD-1+TR group may reflect the state of adaptive immune resistance. In contrast, no significant difference in PD-L2 expression was observed between the Chemo+TR and Anti-PD-1+TR groups (Figure 5e). Regarding the PD-L2 expression on tumor cells and immune cells in the TIME, some studies reported its upregulation after chemotherapy, while others showed no association between PD-L2 expression and chemotherapy [19,20]. Thus, the treatment-related alteration of PD-L2 expression remains controversial.

MHC class II is expressed not only on immune cells, such as dendritic cells and B cells, but also on tumor cells [21]. MHC class II expressed on tumor cells suppresses T-cell function through binding to LAG-3 [17], while it also activates an antitumor immune response by presenting tumor antigens to CD4-positive T cells [22]. We showed that the expression of MHC class II on tumor cells was significantly downregulated after anti-PD-1 therapy compared to chemotherapy (Figure 5f). It was reported that downregulation of MHC class II expression on tumor cells leads to resistance to anti-PD-1 therapy [22]. The downregulation of MHC class II expression on tumor cells after anti-PD-1 therapy in this study may contribute to resistance to anti-PD-1 therapy because it evades LAG-3 axis-mediated immunosuppression but inhibits activation of antitumor immune responses mediated by CD4-positive T cells. Therefore, downregulation of MHC class II expression on tumor cells could be one of the mechanisms underlying resistance to anti-PD-1 therapy.

It was known that CEACAM-1 and CD155 are ligands for TIM-3 and TIGIT, which are major inhibitory immune checkpoint pathways, respectively [23,24]. We previously reported that the expression of CEACAM-1 and CD155 was increased on tumor cells in gastrointestinal metastases of malignant melanoma after anti-PD-1 therapy [14]. In the present study, we showed a trend toward increased expression of CEACAM-1 and CD155 on tumor cells after anti-PD-1 therapy in patients with unresectable advanced or recurrent G/GEJ cancer (*p* = 0.059 and *p* = 0.093, respectively) (Figure 5g,h). Although further investigation with an expanded sample size is necessary, these findings suggest that anti-PD-1 therapy may enhance not only the PD-1 axis but also the TIM-3 and TIGIT axes, potentially leading to greater suppression of CTLs in the TIME in unresectable advanced or recurrent G/GEJ cancer.

In this study, many clusters of CD8-positive T cells, such as those shown in Figure 4, were observed in the Anti-PD-1+TR group, but no significant difference in the number of CD8-positive T cells was observed between the Chemo+TR group and Anti-PD-1+TR group by the evaluation method used in this study (*p* = 0.127) (Figure 5b). Tertiary lymphoid structures (TLS) appear in chronic inflammatory lesions caused by autoimmune diseases and infectious diseases and are also known to exist in the tumor microenvironment of malignant tumors [25]. In addition, it has been reported that patients with malignant tumors accompanied by TLS have a favorable prognosis [26,27]. The clusters of CD8-positive T cells identified in the Anti-PD-1+TR group may also be the outcome of an immune response associated with anti-PD-1 therapy. Future research is expected to elucidate the clinical significance of this finding.

The results of this study demonstrated that, in patients with unresectable advanced or recurrent G/GEJ cancer, the TR group had significantly longer OS compared to the Non-TR group (Figure 3a). In a subgroup of patients who had received anti-PD-1 therapy as well, those who were able to undergo TR also had a significantly better prognosis (Figure 3b). These findings are consistent with trends reported in previous studies [9,10,28]. Furthermore, multivariate analysis in this cohort identified TR after chemotherapy or anti-PD-1 therapy as a significant independent predictor of OS. Based on these results, it is suggested that TR may be a promising treatment strategy when TR is feasible after chemotherapy or anti-PD-1 therapy.

Since this study is a retrospective cohort study, patients assessed as eligible for surgery after chemotherapy were selected for the TR group. In other words, this group consists of a subset of responders with good general condition, and the findings of this study are affected by this selection bias. In addition, due to the small number of cases in this study, it may be difficult to use the findings of this study as a recommended treatment for patients with advanced gastric cancer in clinical practice. Therefore, to establish the clinical significance of TR after chemotherapy or anti-PD-1 therapy in patients with unresectable advanced or recurrent G/GEJ cancer, validation through a prospective randomized controlled trial is desirable.

## 5. Conclusions

This study indicated that immune escape from CTLs may be induced in the TIME in patients with unresectable advanced or recurrent G/GEJ cancer after anti-PD-1 therapy due to the downregulation of HLA class I and MHC class II expression on tumor cells. This observation suggests that these patients may not benefit from immune checkpoint inhibitors anymore, including anti-PD-1 therapy. Moreover, in view of the significantly better prognosis in the Anti-PD-1+TR group, TR may be a promising treatment strategy when TR is feasible after anti-PD-1 therapy. In the current clinical practice in patients with unresectable advanced or recurrent G/GEJ cancer, chemotherapy, which has not been used previously, is often selected as the subsequent treatment after anti-PD-1 therapy. The results of this study, which examined TIME in addition to the clinical perspective, suggested that TR may be a promising treatment strategy when TR is feasible after anti-PD-1 therapy.

## Figures and Tables

**Figure 1 cells-14-01212-f001:**
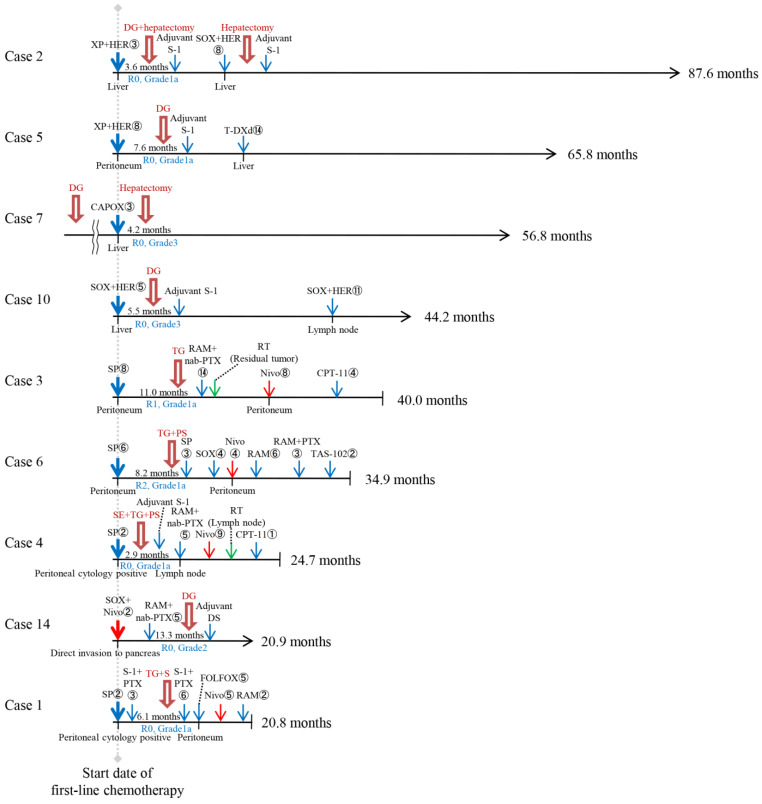
Clinical course of nine patients who underwent tumor resection after chemotherapy. The thick red arrow indicates the surgery. The blue, green, and red arrows indicate chemotherapy, radiotherapy, and immunotherapy, respectively. The numbers listed next to each regimen indicate the number of treatments. The locations of the metastatic tumors are listed below the horizontal axis. The period from the start date of first-line treatment with chemotherapy and/or anti-PD-1 therapy to tumor resection is shown on the horizontal axis. The overall survival is shown on the right side of the horizontal axis. The horizontal line of arrows indicates survivors. CAPOX—capecitabine-oxaliplatin; CPT-11—irinotecan; DG—distal gastrectomy; DS—Docetaxel-S-1; FOLFOX—oxaliplatin-5-fluorouracil-leucovorin-oxaliplatin; HER—trastuzumab; Nivo—nivolumab; PS—pancreato-splenectomy; PTX—paclitaxel; RAM—ramucirumab; RT—radiotherapy; S—splenectomy; SE—subtotal esophagectomy; SOX—S-1-oxaliplatin; SP—S-1-cisplatin; TAS-102—Trifluridine/tipiracil; T-DXd—Trastuzumab deruxtecan; TG—total gastrectomy; XP—capecitabine-cisplatin.

**Figure 2 cells-14-01212-f002:**
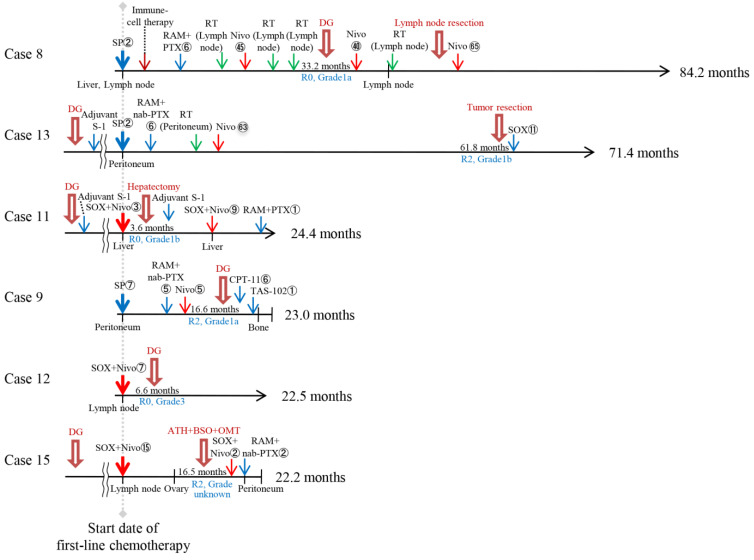
Clinical course of six patients who underwent tumor resection after anti-PD-1 therapy. The thick red arrow indicates the surgery. The blue, green, and red arrows indicate chemotherapy, radiotherapy, and immunotherapy, respectively. The numbers listed next to each regimen indicate the number of treatments. The locations of the metastatic tumors are listed below the horizontal axis. The period from the start date of first-line treatment with chemotherapy and/or anti-PD-1 therapy to tumor resection is shown on the horizontal axis. The overall survival is shown on the right side of the horizontal axis. The horizontal line of arrows indicates survivors. ATH—abdominal total hysterectomy; BSO—bilateral salpingo-oophorectomy; CPT-11—irinotecan; DG—distal gastrectomy; Nivo—nivolumab; OMT—omentectomy; PTX—paclitaxel; RAM—ramucirumab; RT—radiotherapy; SOX—S-1-oxaliplatin; SP—S-1-cisplatin; TAS-102—Trifluridine/tipiracil.

**Figure 3 cells-14-01212-f003:**
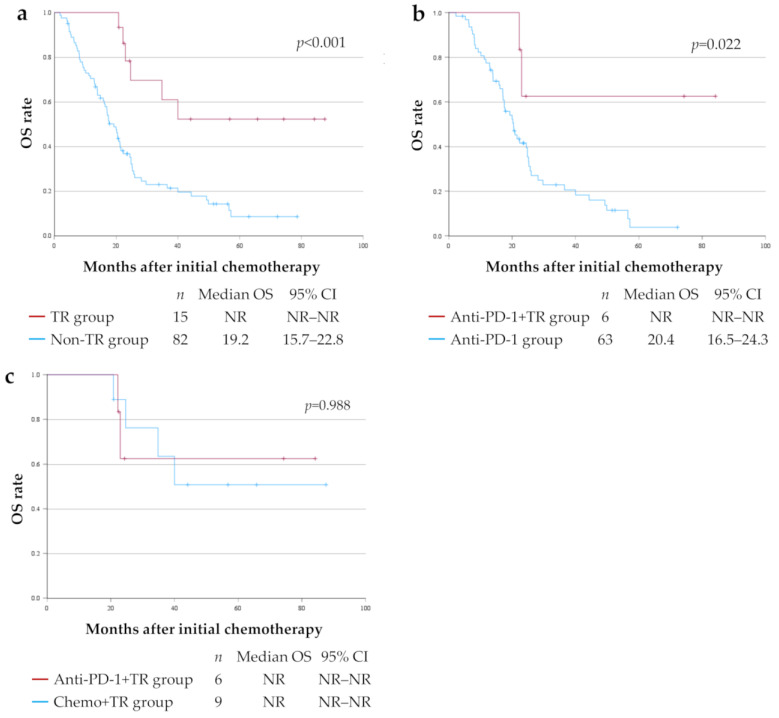
Kaplan–Meier curves for overall survival (OS) for each treatment group. (**a**) OS curves in patients who underwent tumor resection (TR) after chemotherapy or anti-PD-1 therapy (TR group, *n* = 15) and those who underwent chemotherapy and/or anti-PD-1 therapy alone (Non-TR group, *n* = 82). (**b**) OS curves in patients who underwent TR after anti-PD-1 therapy (Anti-PD-1+TR group, *n* = 6) and those who underwent anti-PD-1 therapy in the Non-TR group (Anti-PD-1 group, *n* = 63). (**c**) OS curves in Anti-PD-1+TR group (*n* = 6) and patients who underwent TR after chemotherapy (Chemo+TR group, *n* = 9). CI—confidence interval; PD-1—programmed death 1 receptor; NR—not reached.

**Figure 4 cells-14-01212-f004:**
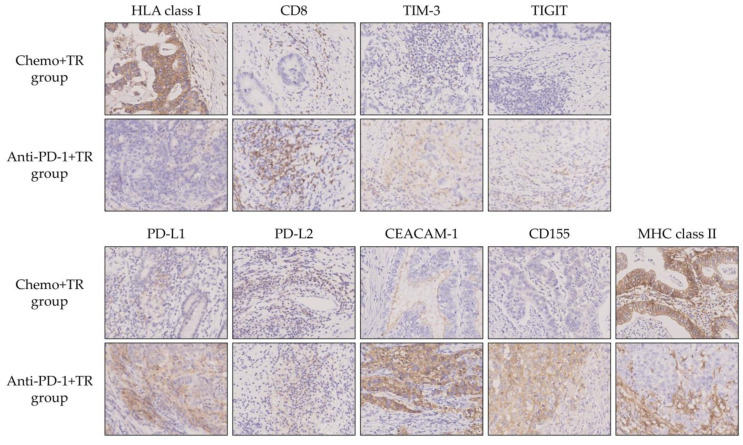
Representative IHC staining with HLA class I, CD8, TIM-3, TIGIT, PD-L1, PD-L2, CEACAM-1, CD155, and MHC class II in patients who underwent tumor resection (TR) after chemotherapy (Chemo+TR group) and those who underwent TR after anti-PD-1 therapy (Anti-PD-1+TR group). Original magnification ×400.

**Figure 5 cells-14-01212-f005:**
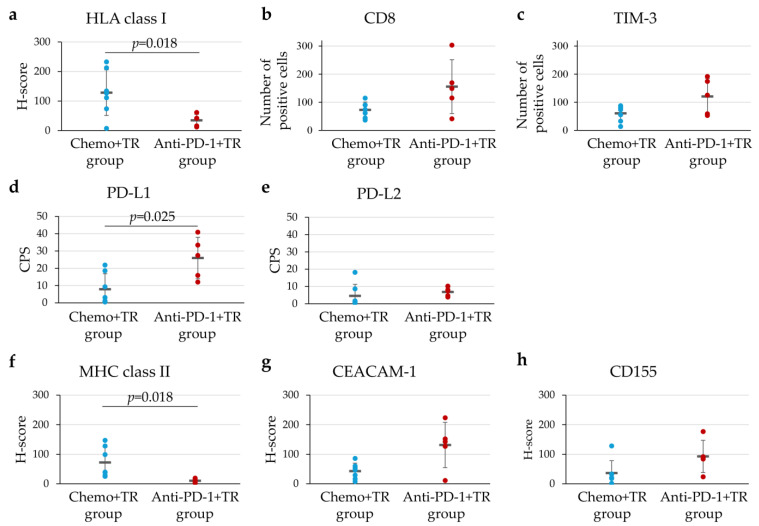
Characteristics of tumor immune microenvironment in patients who underwent tumor resection (TR) after chemotherapy (Chemo+TR group) and those who underwent TR after anti-PD-1 therapy (Anti-PD-1+TR group). Dot plots showing the distribution of H-score, CPS or number of cells for HLA class I (**a**), CD8 (**b**), TIM-3 (**c**), PD-L1 (**d**), PD-L2 (**e**), MHC class II (**f**), CEACAM-1 (**g**), CD155 (**h**) expression evaluated by IHC using surgically resected specimens. Means of H-score, CPS, and number of cells are represented by the horizontal line, together with their error bars representing the standard deviation. Within each molecule category, every dot represents the evaluation of IHC staining in each case.

**Figure 6 cells-14-01212-f006:**
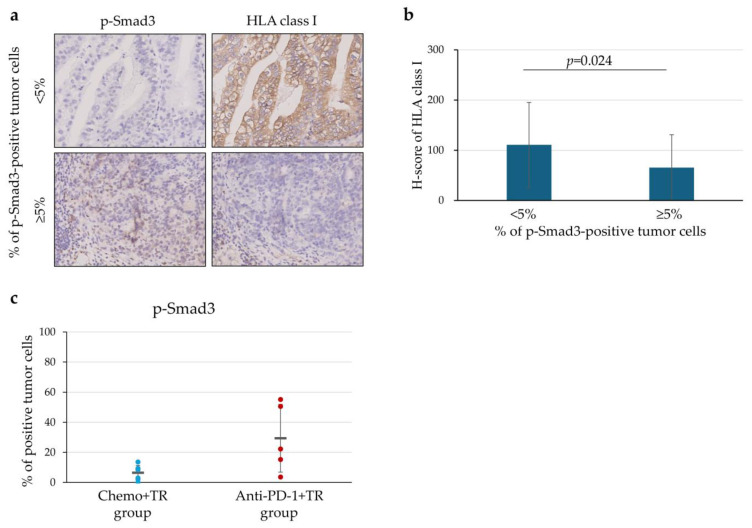
Downregulation of HLA class I expression in relation to p-Smad3 expression. (**a**) Representative IHC staining with p-Smad3 and HLA class I expression between the positive tumor cells for p-Smad3 expression < 5% and ≥5%. The consecutive tissue sections were used in this evaluation. Original magnification ×400. (**b**) Bar graph showing the H score for HLA class I expression level in the group with <5% (30 fields) and the group with ≥5% (30 fields) of tumor cells positive for p-Smad3 expression. The mean percentage is represented by the bars, together with the error bars representing the standard deviation. (**c**) P-Smad3 expression in patients who underwent tumor resection (TR) after chemotherapy (Chemo+TR group) and those who underwent TR after anti-PD-1 therapy (Anti-PD-1+TR group). Dot plots showing the distribution of the percentage of p-Smad-positive tumor cells determined by IHC using surgically resected specimens. The means of the percentage of p-Smad3-positive tumor cells are represented by the horizontal line, together with their error bars representing the standard deviation. Every dot represents the evaluation of IHC staining in one case.

**Table 1 cells-14-01212-t001:** Characteristics and clinical laboratory data of 97 patients.

Characteristics	All Patients (*n* = 97)	Non-TR Group * (*n* = 82)	TR Group * (*n* = 15)	*p*-Value
Age (years), mean ± SD	67.3 ± 10.0	67.8 ± 9.3	64.2 ± 13.1	0.324
Sex				
Male	70 (72.2%)	58 (70.7%)	12 (80.0%)	0.548
Female	27 (27.8%)	24 (29.3%)	3 (20.0%)	
Performance status				
0, 1	95 (97.9%)	80 (97.6%)	15 (100%)	>0.999
2	2 (2.1%)	2 (2.4%)	0 (0%)	
Primary tumor site				
Stomach	93 (95.9%)	79 (96.3%)	14 (93.3%)	0.495
GEJ	4 (4.1%)	3 (3.7%)	1 (6.7%)	
Histological type				
Diffuse	51 (52.6%)	42 (51.2%)	9 (60.0%)	0.667
Non-diffuse	43 (44.3%)	37 (45.1%)	6 (40.0%)	
Unknown	3 (3.1%)	3 (3.7%)	0 (0%)	
HER2 status, positive	10 (10.3%)	7 (8.5%)	3 (20.0%)	0.183
Disease status at first-line chemotherapy				
Unresectable	66 (68.0%)	55 (67.1%)	11 (73.3%)	0.768
Recurrent	31 (32.0%)	27 (32.9%)	4 (26.7%)	
Number of non-curative factors				
1	59 (60.8%)	47 (57.3%)	12 (80.0%)	0.098
≥2	38 (39.2%)	35 (42.7%)	3 (20.0%)	
Number of chemotherapy regimens				
<3	40 (41.2%)	35 (42.7%)	5 (33.3%)	0.499
≥3	57 (58.8%)	47 (57.3%)	10 (66.7%)	
Anti-PD-1 therapy	74 (76.3%)	63 (76.8%)	11 (73.3%)	0.749
Nivolumab	73 (98.6%)	62 (98.4%)	11 (100%)	
Pembrolizumab	1 (1.4%)	1 (1.6%)	0 (0%)	
Laboratory data at first-line chemotherapy				
Alb (g/dL), mean ± SD	3.77 ± 0.52	3.72 ± 0.51	4.05 ± 0.49	0.032
ALP (U/L), mean ± SD	127.2 ± 173.5	135.9 ± 187.4	79.6 ± 19.5	0.010
Cre (mg/dL), mean ± SD	0.861 ± 0.484	0.867 ± 0.522	0.825 ± 0.178	0.568
NLR, mean ± SD	3.14 ± 1.85	3.12 ± 1.86	3.23 ± 0.47	0.842
CEA (ng/mL), mean ± SD	78.58 ± 343.70	86.56 ± 370.57	34.96 ± 114.60	0.310
CA19-9 (U/mL), mean ± SD	1429.20 ± 8028.58	1656.33 ± 8719.19	187.52 ± 435.46	0.134

* Patients underwent tumor resection (TR) after chemotherapy or anti-PD-1 therapy (TR group) and those underwent chemotherapy and/or anti-PD-1 therapy alone (Non-TR group). Alb—albumin; ALP—alkaline phosphatase; CA19-9—carbohydrate antigen 19-9; CEA—carcinoembryonic antigen; Cre—creatinine; GEJ—gastroesophageal junction; HER2—human epidermal growth factor receptor 2; NLR—neutrophil-to-lymphocyte ratio; PD-1—programmed death 1 receptor; SD—standard deviation; TR—tumor resection.

**Table 2 cells-14-01212-t002:** Univariate and multivariate analysis of overall survival in 97 patients treated with chemotherapy and/or anti-PD-1 therapy.

Factors	Category	Univariate Analysis			Multivariate Analysis		
		Hazard Ratio	95% CI	*p*-Value	Hazard Ratio	95% CI	*p*-Value
Age (years)	≥65 (vs. <65)	0.975	0.600–1.585	0.919			
Sex	Male (vs. Female)	0.801	0.476–1.347	0.403			
Performance status	0, 1 (vs. 2)	0.390	0.095–1.610	0.193			
Primary tumor site	Stomach (vs. GEJ)	0.552	0.199–1.527	0.252			
Histological type	Non-diffuse (vs. Diffuse)	0.761	0.478–1.214	0.252			
HER2 status	Positive (vs. Negative)	0.403	0.173–0.935	0.034	0.408	0.169–0.986	0.046
* Disease status	Recurrent (vs. Unresectable)	0.718	0.428–1.204	0.209			
Number of non-curative factors	1 (vs. ≥2)	0.832	0.521–1.328	0.440			
Number of chemotherapy regimens	≥3 (vs. <3)	0.641	0.395–1.042	0.073			
Anti-PD-1 therapy	Yes (vs. No)	1.153	0.647–2.056	0.629			
^#^ TR	Yes (vs. No)	0.258	0.111–0.599	0.002	0.274	0.117–0.643	0.003
Alb (g/dL)	≥3.6 (vs. <3.6)	0.604	0.371–0.985	0.043	0.606	0.377–1.007	0.062
ALP (U/L)	<100 (vs. ≥100)	1.264	0.774–2.065	0.350			
Cre (mg/dL)	<1.0 (vs. ≥1.0)	0.917	0.519–1.621	0.765			
NLR	<3.0 (vs. ≥3.0)	0.608	0.382–0.969	0.036	0.616	0.377–1.007	0.053

* Disease status at first-line treatment with chemotherapy and/or anti-PD-1 therapy (immunotherapy). ^#^ TR after chemotherapy or anti-PD-1 therapy. Alb—albumin; ALP—alkaline phosphatase; CI—confidence interval; Cre—creatinine; GEJ—gastroesophageal junction; HER2—human epidermal growth factor receptor 2; NLR—neutrophil-to-lymphocyte ratio; PD-1—programmed death 1 receptor; TR—tumor resection.

**Table 3 cells-14-01212-t003:** Characteristics of 15 patients who underwent TR after chemotherapy or anti-PD-1 therapy.

Characteristics	All Patients (*n* = 15)	Chemo+TR Group (*n* = 9)	Anti-PD-1+TR Group (*n* = 6)	*p*-Value
Age (years), mean ± SD	64.2 ± 13.1	58.5 ± 13.7	72.8 ± 6.1	0.018
Sex				
Male	12 (80.0%)	7 (77.8%)	5 (83.3%)	>0.999
Female	3 (20.0%)	2 (22.2%)	1 (16.7%)	
Primary tumor site				
Stomach	14 (93.3%)	8 (88.9%)	6 (100%)	>0.999
GEJ	1 (6.7%)	1 (11.1%)	0 (0%)	
Histological type				
Diffuse	6 (40.0%)	4 (44.4%)	2 (33.3%)	>0.999
Non-diffuse	9 (60.0%)	5 (55.6%)	4 (66.7%)	
HER2 status, positive	3 (20.0%)	3 (33.3%)	0 (0%)	0.229
Disease status at first-line chemotherapy				
Unresectable	11 (73.3%)	8 (88.9%)	3 (50.0%)	0.235
Recurrent	4 (26.7%)	1 (11.1%)	3 (50.0%)	
Number of non-curative factors				
1	12 (80.0%)	9 (100%)	3 (50.0%)	0.044
≥2	3 (20.0%)	0 (0%)	3 (50.0%)	
Residual tumor status				
R0	10 (66.7%)	7 (77.8%)	3 (50.0%)	0.329
R1, R2	5 (33.3%)	2 (22.2%)	3 (50.0%)	
Pathological tumor response				
Grade1a	8 (53.3%)	6 (66.7%)	2 (33.3%)	0.201
Grade1b	2 (13.3%)	0 (0%)	2 (33.3%)	
Grade2	1 (6.7%)	1 (11.1%)	0 (0%)	
Grade3	3 (20.0%)	2 (22.2%)	1 (16.7%)	
Unknown	1 (6.7%)	0 (0%)	1 (16.7%)	

Anti-PD-1+TR—tumor resection after anti-PD-1 therapy; Chemo+TR—tumor resection after chemotherapy; GEJ—gastroesophageal junction; HER2—human epidermal growth factor receptor 2; PD-1—programmed death 1 receptor; SD—standard deviation; TR—tumor resection.

## Data Availability

The datasets generated and/or analyzed during the current study are available from the corresponding author on reasonable request.

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
