# Peer review of "Residual Tumor Resection After Anti-PD-1 Therapy: A Promising Treatment Strategy for Overcoming Immune Evasive Phenotype Induced by Anti-PD-1 Therapy in Gastric Cancer"

_cells, 2025, doi:10.3390/cells14151212_

Round 1

Reviewer 1 Report

Comments and Suggestions for Authors

This article titled “Residual tumor after anti-PD-1 therapy in gastric cancer showed immune evasion phenotype including HLA class I downregulation” has a significant result sections (3.1, 3.2, 3.4) demonstrating that tumour resection (TR) after first-line chemotherapy and/or immunotherapy (anti-PD-1 treatment) extended the overall survival (OS) of patients compared to those patients who only had chemotherapy and/or immunotherapy without subsequent TR. Therefore, they concluded that TR was approved as a promising treatment option for patients with unresectable advanced or recurrent gastric/gastroesophageal junction cancers. Afterwards, the authors added two result sections (3.3, 3.5, 3.6) to further compare TR after chemotherapy and immunotherapy in terms of tumour immune evasion. They found that anti-PD-1 treatment induced downregulation of HLA-I and MHC-II, which contributed to the tumour's immune evasion. The title of this article may need to be changed to reflect the focus of this study, as half of the results demonstrated the advantage of TR after first-line treatment over chemo- and/or immunotherapies alone.

When comparing the TR after chemotherapy with the TR after anti-PD-1 treatment (immunotherapy), it is better to compare the overall survival of patients who had TR after different first-line therapies as well.

Anti-PD-1 therapy is an immunotherapy that should not be included in chemotherapy. This should be corrected in the text, Table 1, Figure 2, and Figure 3a (better be “TR after first-line chemotherapy and/or immunotherapy”.

Likewise, the title “3.5. Characteristics of TIME in unresectable advanced or recurrent G/GEJ cancer treated with chemotherapy” should be changed to “Characteristics of TIME in unresectable advanced or recurrent G/GEJ cancer treated with chemotherapy and immunotherapy”.

How can the authors be sure that HLA-I, MHC-II and p-Smad3 were stained on tumour cells rather than the tumour stromal cells or other non-tumour cells within the tumour microenvironment?

The pictures in Fig.4 showed that the expression of CD8 and CEACAM-1 was increased in the anti-PD-1-treated group. Please comment on this.

Comments on the Quality of English Language

The English could be improved to express the research work more clearly.

Author Response

This article titled “Residual tumor after anti-PD-1 therapy in gastric cancer showed immune evasion phenotype including HLA class I downregulation” has a significant result sections (3.1, 3.2, 3.4) demonstrating that tumour resection (TR) after first-line chemotherapy and/or immunotherapy (anti-PD-1 treatment) extended the overall survival (OS) of patients compared to those patients who only had chemotherapy and/or immunotherapy without subsequent TR. Therefore, they concluded that TR was approved as a promising treatment option for patients with unresectable advanced or recurrent gastric/gastroesophageal junction cancers. Afterwards, the authors added two result sections (3.3, 3.5, 3.6) to further compare TR after chemotherapy and immunotherapy in terms of tumour immune evasion. They found that anti-PD-1 treatment induced downregulation of HLA-I and MHC-II, which contributed to the tumour's immune evasion. The title of this article may need to be changed to reflect the focus of this study, as half of the results demonstrated the advantage of TR after first-line treatment over chemo- and/or immunotherapies alone.

Response

Thank you for this suggestion. We changed the title as follows: Therapeutic potential of tumor resection in gastric cancer patients after anti-PD-1 therapy: due to immune evasion phenotype including HLA class I downregulation.

When comparing the TR after chemotherapy with the TR after anti-PD-1 treatment (immunotherapy), it is better to compare the overall survival of patients who had TR after different first-line therapies as well.

Response

We added OS curves in nine patients who underwent TR after chemotherapy (Chemo+TR group) and six patients who underwent TR after anti-PD-1 therapy (Anti-PD-1+TR group) as Figure 3c. This result is described in lines 250-251.

Anti-PD-1 therapy is an immunotherapy that should not be included in chemotherapy. This should be corrected in the text, Table 1, Figure 2, and Figure 3a (better be “TR after first-line chemotherapy and/or immunotherapy”.

Response

We have carefully revised the text, Tables and Figures to clearly distinguish between immunotherapy (anti-PD-1 therapy) and chemotherapy.

Likewise, the title “3.5. Characteristics of TIME in unresectable advanced or recurrent G/GEJ cancer treated with chemotherapy” should be changed to “Characteristics of TIME in unresectable advanced or recurrent G/GEJ cancer treated with chemotherapy and immunotherapy”.

Response

We have done it.

How can the authors be sure that HLA-I, MHC-II and p-Smad3 were stained on tumour cells rather than the tumour stromal cells or other non-tumour cells within the tumour microenvironment?

Response

In this study, we identified tumor lesions by referring to hematoxylin-eosin stained sections of consecutive tissue sections and then evaluated each staining on tumor cells. We added this comment in lines 129-131.

The pictures in Fig.4 showed that the expression of CD8 and CEACAM-1 was increased in the anti-PD-1-treated group. Please comment on this.

Response

Regarding the number of CD8-positive T cells, we added references (#25-27) and comments in lines 393-403. We described comments about CEACAM-1 in lines 384-394.

Reviewer 2 Report

Comments and Suggestions for Authors

The authors demonstrated that patients undergoing anti-PD-1 therapy experienced adverse effects within the tumor immune microenvironment (TIME), including the downregulation of MHC class I and II and the upregulation of CEACAM-1 and CD155, ultimately contributing to the conversion of a “hot” tumor into a “cold” tumor. Importantly, they found that the anti-PD-1 therapy–induced downregulation of MHC class I and II was associated with activation of the TGF-β signaling pathway.

Overall, the study is well-designed and straightforward. These findings will be valuable for establishing clinical therapeutic strategies for gastric cancer.

The reviewer suggested minor revisions:

  1. The resolution of the figures and graphs is poor and should be improved.
  2. A simple table of contents (TOC) would be helpful for readers’ understanding.

Author Response

The authors demonstrated that patients undergoing anti-PD-1 therapy experienced adverse effects within the tumor immune microenvironment (TIME), including the downregulation of MHC class I and II and the upregulation of CEACAM-1 and CD155, ultimately contributing to the conversion of a “hot” tumor into a “cold” tumor. Importantly, they found that the anti-PD-1 therapy–induced downregulation of MHC class I and II was associated with activation of the TGF-β signaling pathway.

Overall, the study is well-designed and straightforward. These findings will be valuable for establishing clinical therapeutic strategies for gastric cancer.

The reviewer suggested minor revisions:

The resolution of the figures and graphs is poor and should be improved.

Response

We thank the reviewer for the favorable evaluation of our manuscript. We have tried to improve the resolution of the figures and graphs.

A simple table of contents (TOC) would be helpful for readers’ understanding.

Response

We added a simple TOC in lines 152-154.

Reviewer 3 Report

Comments and Suggestions for Authors

The article presents a fundamental piece of research that delves into the mechanisms of immune evasion in gastric cancer after anti-PD-1 therapy. The study highlights the downregulation of HLA class I, a key factor in the tumor's ability to evade immune detection. The research provides valuable insights into the molecular mechanisms of immune evasion, which is crucial for developing more effective treatment strategies. The study contributes to the understanding of the challenges posed by residual tumors after anti-PD-1 therapy.

The design of the study could be more robust. Clear criteria for including and excluding patients from the study would enhance the generalizability and reproducibility of the findings. The practical significance of the research is not fully explored. The article does not provide clear guidelines or recommendations for clinical practice based on the findings. The lack of specific criteria for evaluating the practical significance limits the potential impact of the research on clinical decision-making. Overall, the article presents a promising piece of research with significant potential for advancing the field. However, to fully realize its impact, the study would benefit from more detailed design and clearer articulation of its practical implications.

Author Response

The article presents a fundamental piece of research that delves into the mechanisms of immune evasion in gastric cancer after anti-PD-1 therapy. The study highlights the downregulation of HLA class I, a key factor in the tumor's ability to evade immune detection. The research provides valuable insights into the molecular mechanisms of immune evasion, which is crucial for developing more effective treatment strategies. The study contributes to the understanding of the challenges posed by residual tumors after anti-PD-1 therapy.

The design of the study could be more robust. Clear criteria for including and excluding patients from the study would enhance the generalizability and reproducibility of the findings. The practical significance of the research is not fully explored. The article does not provide clear guidelines or recommendations for clinical practice based on the findings. The lack of specific criteria for evaluating the practical significance limits the potential impact of the research on clinical decision-making. Overall, the article presents a promising piece of research with significant potential for advancing the field. However, to fully realize its impact, the study would benefit from more detailed design and clearer articulation of its practical implications.

appropriateness of these citations.

Response

We thank the reviewer for the positive evaluation of our manuscript. At first, we added information regarding the eligibility criteria and exclusion criteria in this study in lines 82-89.

Regarding the reviewers’ comments on guidelines or recommendations for clinical practice based on the findings, we wanted to emphasize our findings as recommendations for clinical practice. However, the number of cases in this study is small, and the precision of the analysis results may not be sufficient. We added this limitation in lines 418-420.

Round 2

Reviewer 1 Report

Comments and Suggestions for Authors

The authors addressed my concerns properly. However, the title of the paper can be further improved to better express the focus of the paper.

Author Response

Comments to Author:

Reviewer 1

The authors addressed my concerns properly. However, the title of the paper can be further improved to better express the focus of the paper.

Response

We thank the reviewer for the favorable evaluation of our manuscript and this suggestion. We changed the title again as follows: Residual Tumor Resection After Anti-PD-1 Therapy: A Promising Treatment Strategy for Overcoming Immune Evasive Phenotype induced by Anti-PD-1 Therapy in Gastric Cancer.